# Identifying Populations at Risk for Lung Cancer Mortality from the National Health and Nutrition Examination Survey (2001–2018) Using the 2021 USPSTF Screening Guidelines

**DOI:** 10.3390/ijerph21060781

**Published:** 2024-06-15

**Authors:** Vivian Tieu, Samuel MacDowell, Sedra Tibi, Bradley Ventayen, Mukesh Agarwal

**Affiliations:** Department of Medical Education, Faculty of Medicine, California University of Science and Medicine, Colton, CA 92324, USA; samuel.macdowell@md.cusm.edu (S.M.); sedra.tibi@md.cusm.edu (S.T.); bradley.ventayen@md.cusm.edu (B.V.); mukesh.agarwal@cusm.edu (M.A.)

**Keywords:** lung cancer screening, preventative health, health inequity, race, socioeconomic

## Abstract

Lung cancer (LC) is the leading cause of cancer mortality in the United States. To combat this predicament, early screening and critically assessing its risk factors remain crucial. The aim of this study was to identify the value of specific factors from the National Health and Nutrition Examination Survey (NHANES) from 2001–2018, as they relate to lung cancer mortality in the US Preventive Services Task Force (USPSTF)-eligible population. A total of 3545 adults who met USPSTF criteria were extracted from 81,595 NHANES participants. The LC Death Risk Assessment Tool was used to calculate the number of deaths per 1000 individuals. The Mann–Whitney U test and one-way ANOVA determined the statistical significance of the factors involved in LC mortality. Male sex, African and Hispanic ethnicity, lower education attainment, and secondhand exposure to cigarette smoke correlated with an increased risk of LC mortality. Additionally, the factor of emotional support from NHANES data was analyzed and did not show any benefit to reducing risk. By identifying individuals at high-risk, preventative measures can be maximized to produce the best possible outcome

## 1. Introduction

Cancer is the second leading cause of death in the United States [1]. In 2024, approximately 1.95 million people will be diagnosed with cancer, of whom about 609,820 (31.2%) individuals will die [2]. Lung cancer (LC) is the leading cause of cancer mortality in both men and women because, at diagnosis, most patients are usually already at an advanced stage. In addition, racial and social disparities limit healthcare access, impacting prognosis and survival [3]. To decrease mortality, early screening and preventive measures are crucial for those at risk.

Early trials examining chest radiographs and sputum cytology as screening tools for lung cancer in cigarette smokers failed to demonstrate a reduction in lung cancer mortality [4]. However, the 2011 Dutch–Belgian Randomized Lung Cancer Screening Trial (Dutch acronym NELSON) and the 2013 National Lung Screening Trial (NLST) studies both demonstrated significant mortality benefits of low-dose computed tomography (LDCT) screening [5,6]. In 2013, based on findings from these studies, the US Preventive Services Task Force (USPSTF) recommended annual LDCT lung screening for all individuals aged 55–80 years with a 30 pack-year smoking history (current smokers or smokers who quit for less than 15 years) [7]. In 2021, after reviewing four lung cancer history models, the USPSTF modified its screening guidelines by reducing both the lower age and smoking history to 50 years and 20 pack-years, respectively [8]. In addition to contributing to the USPSTF guidelines for LDCT lung cancer screening, the NLST study helped to develop the NIH’s Risk-based NLST Outcomes Tool (RNOT), which utilizes a risk-based algorithm for the probability of LC diagnosis and death, with and without LC screening [9]. The importance of this tool is that utilizing a risk-based model, as opposed to solely following blanket USPSTF recommendations, may allow for increased screen-avertable LC deaths [10]. However, earlier screening is not without problems, and it may cause increased false-positive results and radiation-related deaths.

Many programs in the United States have incorporated the USPSTF lung cancer screening guidelines, notably State Medicaid Programs. As of October 2023, over twenty states offer coverage for lung cancer screening, according to the updated USPSTF 2021 guidelines. A majority of the remaining states offer different types of coverage for screenings, either because they follow outdated USPSTF or non-USPSTF guidelines [11]. The World Trade Center Health Program offers lung cancer screening according to the updated USPSTF guidelines as a medical benefit to its members [12]. Prominent institutions have established programs to advocate for lung cancer screening according to the updated USPSTF 2021 guidelines, such as the Smilow Cancer Hospital Lung Cancer Screening Program, part of Yale Medicine, and the UCSF Lung Cancer Screening Program [13,14]. However, despite spreading availability for LDCT cancer screening, certain populations remain at higher risk for lung cancer mortality.

LC is a multifactorial disorder; the risk factors are categorized as non-modifiable (e.g., age, sex, ethnicity) or modifiable (e.g., smoking history, occupation). Several studies have demonstrated that patients from minority communities (e.g., African Americans, Hispanics) have a higher incidence and mortality compared to their White counterparts, and they are less likely to undergo LC screening [3], the causes of which have been attributed to fatalism, medical mistrust, and bias against surgery [15,16]. Furthermore, there are racial and social disparities with LC screening, diagnosis, and treatment. As a result, it is crucial that the USPSTF guidelines and RNOT assessment address socioeconomic and cultural factors so that they are applicable to most, if not all, patient populations in the United States. The goal of this study was to analyze the updated 2021 USPSTF guidelines using the National Health and Nutrition Examination Survey (NHANES) database from 2001–2018 to characterize the patient population eligible to undergo LC screening. It also aimed to identify and further explore the multifactorial and complex factors contributing to LC development and mortality.

## 2. Materials and Methods

The NHANES is a biennial study run by the National Center for Health Statistics under the auspices of the CDC. It is a dataset that roughly models a representative population makeup of the United States across multiple demographics, namely, age, ethnicity, and gender with distribution among 15 selected counties within the United States [17]. It encompasses at least 5000 participants during each 2-year cycle. Data from the NHANES dates to 1960; however, collection for the purpose of this study necessitated that a cross-sectional retrospective cohort study analysis from the NHANES database was limited to the years of 2001–2018; furthermore, many cycles from earlier datasets did not collect adequate information needed for this study. Specifically, many surveys did not ask questions with regards to concomitant lung pathology and smoking to accurately calculate lung cancer risk.

Between these years, a total of 81,595 participants were identified, who were further filtered by the inclusion and exclusion criteria to meet the USPSTF lung screening recommendations. The inclusion criteria were (a) participants aged 50–80 years and (b) smokers with at least a 20 pack-year smoking history who were actively smoking or quit in the last 15 years. Exclusion criteria consisted of participants with (a) a concomitant LC history, (b) ages below 50 and above 80, and (c) inconsistent histories (e.g., reported time for smoking cessation exceeded total smoking time). Out of the initial 81,595 subjects, 3545 participants met the criteria (Figure 1). A breakdown of the 2021 USPSTF-eligible lung cancer screening population characteristics from the NHANES (2001–2018) are detailed in Table 1.

These NHANES participants were used to analyze the effects of the following: gender, race, education, secondhand smoke exposure, and emotional support. Of these 3545 participants, only 1905 participants answered the NHANES survey question pertaining to secondhand smoke exposure, and only 1176 participants answered the question pertaining to emotional support. Each participant’s survey data were entered into the Lung Cancer Death Risk Assessment Tool (LCDRAT) of the National Cancer Institute’s Division of Cancer Epidemiology & Genetics [18]. LCDRAT uses variables developed from various lung cancer incidence and mortality models from 2003–2018. Those variables and subsequent data were extrapolated from the NHANES dataset (2001–2018), as depicted in Table 2. It is important to mention that although LCDRAT asks for family history of lung cancer, it is not an available variable asked within the NHANES. As a result, all participants were assumed to have no lung cancer genetic risk factors. The LCDRAT value determines the number of individuals who would die in the next 5 years without LC screenings (per 1000 individuals); a higher LCDRAT value signifies a higher risk for LC death. For gender, emotional support, and secondhand smokers, nonparametric statistical analysis on the frequency of LCDRAT values was calculated using the Mann–Whitney U test. For race and education, the nonparametric statistical analysis on the frequency of LCDRAT values was calculated using the one-way ANOVA on Ranks test, and a post hoc analysis was conducted on the results. For race, Bonferroni correction was used to adjust significance values to reduce type 1 errors. For education, the Benjamini–Hochberg method was used to adjust significance values to reduce type 1 errors. The single value median represented the center of data distribution.

## 3. Results

### 3.1. Risk of Lung Cancer Death by Gender

A total of 2271 males and 1274 females in the NHANES database (2001–2018) qualified for USPSTF screening guidelines for LC, regardless of screening, with more men meeting screening guidelines, resulting in a 2:1 ratio of males–females. These patients were stratified to determine frequencies of males and females in each risk group of LC death (defined as the number of LC deaths per 1000). The risk for LC deaths for males and females was 15.84 and 11.95 LC deaths per 1000, respectively. (U-value = 1,261,714.000, *p* < 0.001) (Figure 2).

### 3.2. Risk of Lung Cancer Death by Race

Of the 3545 cases qualifying for USPSTF screening guidelines for LC, regardless of screening, the NHANES database from 2001–2018 identified 2017 Non-Hispanic Whites, 801 African Americans, 463 Hispanics, and 264 of “Other Race”. The median for Non-Hispanic White, African Americans, Hispanics, and “Other Race” for LC deaths was 15.21, 18.35, 9.06, and 7.97 per 1000, respectively. There was a statistically significant difference in the risk stratification for LC death between these three racial categories (H-value = 119.227, *p* < 0.001). The post hoc test for one-way ANOVA analysis, used to perform pairwise comparisons, showed the statistical significance between the various groups as follows: Non-Hispanic White and Hispanic (*p* < 0.001), Non-Hispanic White and African American (*p* = 0.013), and African American and Hispanic (*p* < 0.001). There was no statistical significance between Hispanic and “other race” (*p* = 0.951). See Figure 3, Table 3 for more information.

### 3.3. Risk of Lung Cancer Death by Education Attainment Level

The NHANES database identified 3545 cases qualifying for USPSTF screening guidelines for LC from 2001–2018. The breakdown for the highest level of education attainment and their respective number of cases is as follows: 440 achieving an education level of less than 9th grade, 725 reaching 9th–12th grade, 984 being a high school graduate or Graduate Equivalency Degree (GED), 1031 having some college education or an Associates of Arts (AA) degree, and 364 being a college graduate or above. The median for <9th grade, 9th–12th grade, high school graduate or GED, and college education/AA degree was 22.80, 17.4, 13.69, and 11.44 LC deaths per 1000, respectively. There was a statistically significant difference in the risk stratification for LC death between these five education levels (H-value = 116.711, *p* < 0.001). The post hoc test for one-way ANOVA analysis was used to perform pairwise comparisons, with findings of statistical significance between <9th grade and 9th–12th grade (*p* = 0.006), 9th–12th grade and high school graduate/GED (*p* < 0.001), and high school graduate/GED and some college/AA degree (*p* < 0.001). There was no statistical significance between some college/AA degree and college graduate and above (*p* = 0.586). (Figure 4, Table 3).

### 3.4. Risk of Lung Cancer Death by Presence of Emotional Support

The NHANES database includes questions regarding the presence of emotional support, leaving the definition broad and up to the discretion of the participant. The questions regarding emotional support were asked in the following English text: “Now I would like to ask a few questions about {your/SP’s} friends and family. Can {you/SP} count on anyone to provide {you/him/her} with emotional support such as talking over problems or helping {you/him/her} make a difficult decision?” [19]. Of the 3545 participants who qualified for USPSTF screening guidelines for LC from 2001–2018, 1176 answered this survey question, with 1066 participants answering “Yes” to denote the presence of emotional support, and 110 participants answering “No” to indicate the absence of emotional support. The mean rank for those who answered “Yes” resulted in a risk of 585.17, and the mean rank for those answering “No” resulted in a risk of 620.80. The Mann–Whitney U test showed no significant difference in the risk stratification for LC death between those indicating the presence of emotional support and those indicating no emotional support (U-value = 55,077.00, *p* = 0.295) (Figure 5).

### 3.5. Risk of Lung Cancer Death by Secondhand Smoke Exposure

The NHANES database includes questions regarding chronic secondhand smoke exposure, phrased as the presence of smoking individuals at their place of residence. Of the 3545 participants who qualified for USPSTF screening guidelines for LC from 2001–2018, 1905 answered this survey question, with 998 participants answering “Yes” to indicate secondhand smoke exposure, and 907 participants answering “No” to indicate the absence of secondhand smoke exposure. These cases were stratified to determine frequencies of participants in each risk group of LC death, which is defined as the number of LC deaths per 1000. The central tendency for each group was established as the median, a single value representing the center of data distribution. The median for those indicating “Yes” resulted in a risk of 17.00 LC deaths per 1000, and the median for those indicating “No” resulted in 10.97 LC deaths per 1000. Using the Mann–Whitney U test to evaluate for significance, there was a statistically significant difference in the risk stratification for LC death between those exposed to secondhand smoke and those not exposed (U-value = 342,346.000, *p* < 0.001) (Figure 6).

## 4. Discussion

For most of the 20th century, LC has remained the cancer with the highest mortality rate for both males and females. Screening for lung cancer is relatively new, with the recent USPSTF adoption of new guidelines in 2021, which is an update from their first-time adoption in 2013. Mortality benefit is associated with early lung cancer diagnosis, with both NELSON and NLST studies demonstrating a 20% reduction in mortality through the use of annual LDCT; in some cases, mortality was reduced by 39%, depending on the population studied [15]. To understand the potential effect of LDCT screening, it is important to characterize populations most at risk. To the best of our knowledge, this analysis is the first to utilize the NHANES database to identify the characteristics associated with increased LC mortality risk, as per the updated 2021 USPSTF screening guidelines. It provides significant insight into the institutional, racial, cultural, and social factors contributing to LC risk and mortality that may account for discrepancies in LC survival from 2001–2018.

Several studies have shown that females have better cancer outcomes than males [20,21,22,23,24]. One hypothesis suggests these sex differences in LC survival are attributed to females’ relatively better response to LC treatment, earlier diagnosis, and lower comorbidity [25]. Several studies found that females had a higher rate of healthcare access and preventative medical care service utilization, which may be attributed to gender shaping one’s perspective on their identity and health [21,22,23]. Notably, another study concluded that men were less likely to undergo blood pressure and cholesterol screening, suggesting a correlation with traditional masculinity norms [24]. Social gender constructs and the expectation to maintain masculinity encourages males to become excessively self-reliant, often contributing to poor health behaviors and manifesting in healthcare delays, noncompliance with physician recommendations, and unwillingness to undergo preventative health measures, such as recommended USPSTF LC screening [21,22,23,24,25].

Underutilization of healthcare services may also explain the statistically significant difference in LC mortality risk between African American, Non-Hispanic White, and Hispanic subjects identified in our study. Carter-Harris et al. showcased an inverse relationship between medical mistrust and patient willingness to undergo LC screening [26]. African American and Hispanic populations reported higher levels of medical mistrust compared to their White counterparts [3,24,27]. Medical mistrust amongst African American patients can be attributed to historical experiences (e.g., Tuskegee syphilis study) and structural racism within the United States [24,28]. Lin et al. also revealed that African Americans were more likely than White and other minority populations to exhibit negative surgical beliefs, fatalism, and medical mistrust, contributing to discrepancies in LC treatment and screening [15,29]. Notably, there is a positive correlation between healthcare provider recommendations and patient participation in LC screening [26]. When discussing the value of lung cancer screenings, this study highlights the importance of combating medical mistrust, such as using educational outreach to promote screening policies and addressing the reason for patient hesitancy to object towards care [30,31].

Additionally, socioeconomic (SES) factors further contribute to lower rates of healthcare access, participation, and medical compliance amongst African American and other minority patient populations [32]. Those with a lower SES status are more likely to be diagnosed with advanced stage LC with a consequently poorer prognosis, and African American patients have consistently experienced relatively more advanced stage cancer diagnoses compared to their White counterparts [33]. Consistent with findings in our analysis, previous studies also demonstrate an increased risk of LC mortality in those with lower educational levels [34,35]. These findings may be attributed to an increased understanding of the benefits of preventative health measures such as early screening, no smoking, active lifestyle, and healthy diet [30,36]. As observed in our study, secondhand smoking exposure in childhood increases risk of LC later in life [37]. An interesting consideration is that children and adults belonging to a lower SES status, regardless of their own smoking status, are more likely to be exposed to secondhand smoke at home, their respective occupations, or in utero [38,39]. In addition to secondhand smoke exposure in the home, occupational exposures may also contribute to lung cancer. One study demonstrated that diesel exhaust exposure increased squamous cell and small cell lung carcinomas risk in men [40]. Another observed the increased risk of all major histologic types of LC in workers exposed to crystalline silica, such as men and women employed in manufacturing, construction, and sandblasting [40,41]. These related factors may contribute to the racial and SES trends observed in LC morbidity and mortality.

Healthcare insurance serves as an additional determinant of LC risk and outcomes. As the United States does not offer its citizens universal health care coverage, individuals stemming from traditionally underserved communities (e.g., people of color, lower SES, etc.) are more likely to have poorer health outcomes [42]. Despite the passage of the Affordable Care Act and subsequent expansion in 2014, many Americans continue to face health care inequities [43]. One study demonstrated that although increased Medicare access resulted in increased LC screening among high-risk men, many continued to be uninsured and were unable to receive adequate screening [44]. In addition, uninsured and Medicaid LC patients also failed to receive appropriate chemotherapy [45]. However, the further incorporation of value-based care models and pay for performance (P4P) may help alleviate the discrepancies amongst different insurance holders. Although many systematic reviews have found little evidence that P4P improves patient outcomes, it has shown improvements in health processes, particularly screening rates [46]. Value-based care models and P4P studies have not been attempted with lung cancer screening using LDCT, however, it has been widely shown to be very underutilized for screening [47,48]. As cancer is a time-sensitive disease, highly dependent on age-appropriate screening and early initiation of treatment, it is evident that healthcare coverage and utilization of P4P analogous models play a significant role in LC outcomes.

In this study, we identified groups at increased risk of LC mortality, including those who are male, African American, of a lower educational level, and exposed to secondhand smoke. These groups should be most encouraged to follow the 2021 USPSTF guidelines for LC screening. In addition to the role of institutional, SES, and cultural factors in LC morbidity and mortality explored in this study, it may be worthwhile to expand upon the USPSTF guidelines to evaluate other risk prediction models for LC [49]. Moreover, investigations regarding the role E-cigarettes or other non-nicotine vaping products play in LC risk, especially in younger patients, may improve the current screening guidelines [50,51].

### Strengths and Limitations

There are several limitations to note. The NHANES dataset encompasses a representation of the US population over nearly two decades, which is likely to have changed with migration. Additionally, the subjectivity of the survey questions provided to participants regarding emotional support may diminish the objectivity and skew the results. Furthermore, there was limited information available regarding other SES variables (e.g., income and occupation), races/ethnicities that were not White, African American, and Hispanic, and occupational exposures related to lung cancer. Another area of bias is that the LCDRAT calculation relies on lung cancer family history, a variable that the NHANES does not track. Although smoking is the leading attributable risk factor for lung cancer, accounting for up to 90% of global cases, genetic risk factors cannot be ignored, and it is much higher in certain populations [52]. Other potential biases include recall bias in former smokers, which is reflected by a patient’s tendency to associate good outcomes with their quitting, and may, in fact, overestimate their total quit years. The NHANES also relies heavily on patient history, rather than medical records or spirometry, making it challenging for all participants to know if they in fact have concomitant lung conditions.

## 5. Conclusions

This study identified the patient population eligible for LC screening as per the 2021 USPSTF guidelines, using the NHANES database. However, since smoking is not the sole contributor to lung cancer, it is obligatory to address societal, cultural, and economic variables that increase the risk (e.g., male sex, African American ethnicity, and lower education, in this study). Thus, by identifying individuals needing an additional focus, we shall add to winning our constant battle against lung cancer.

## Figures and Tables

**Figure 1 ijerph-21-00781-f001:**
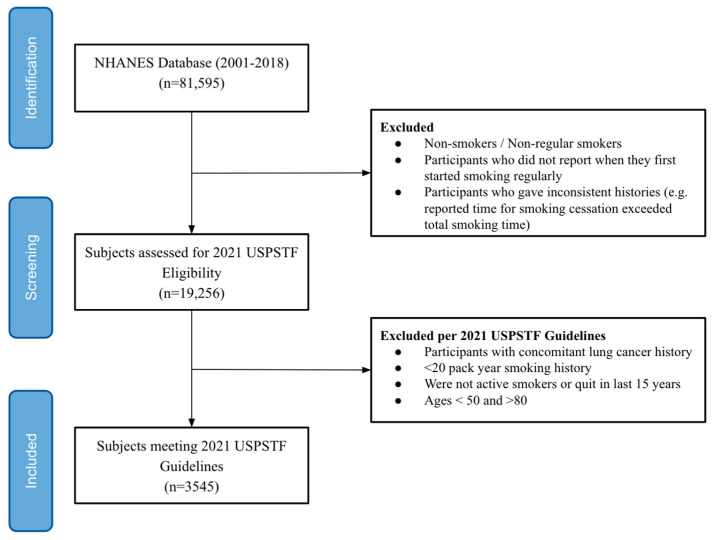
CONSORT Flow Diagram for Participant Selection. From 2001–2018, the NHANES had 81,595 participants available for data screening. Subjects were excluded if they were non-smokers, non-regular smokers, did not report when they first started smoking, and had inconsistent histories. This yielded 19,256 subjects. Out of the 19,256, a total of 3545 met 2021 USPSTF lung cancer screening guidelines and were included in the final analysis of this study.

**Figure 2 ijerph-21-00781-f002:**
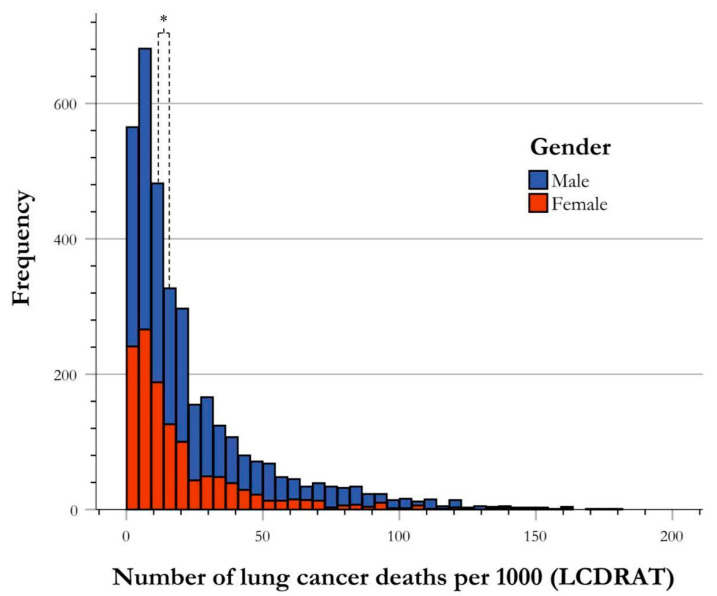
Frequency distribution of risk of lung cancer death by gender. The risk stratification of LC deaths by gender illustrated by frequency of USPSTF-qualified cases of LC deaths per 1000. The central tendency of data for each gender group is established as the median, and is indicated in the figure correlating with group color, with respect to the figure key. The medians of male and female for risk stratification for LC death were found to be significantly different using Mann–Whitney U analysis, as shown using an asterisk (*).

**Figure 3 ijerph-21-00781-f003:**
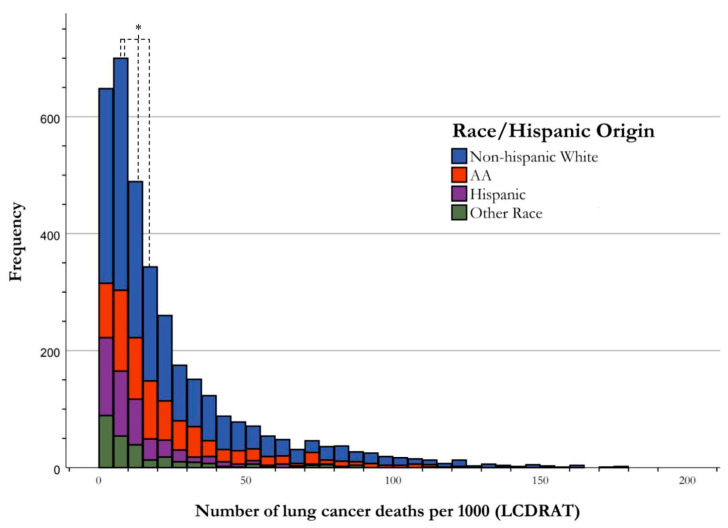
Frequency distribution of risk of lung cancer death by race. Risk stratification of LC deaths by race illustrated by frequency of USPSTF-qualified cases of LC deaths per 1000. The central tendency of data for each racial group is established as the median, and is indicated in the figure correlating with group color, with respect to the figure key. The medians of Non-Hispanic White, African American, Hispanic, and “Other Race” groups for risk stratification for LC death were found to be significantly different using one-way ANOVA analysis, as shown using an asterisk (*).

**Figure 4 ijerph-21-00781-f004:**
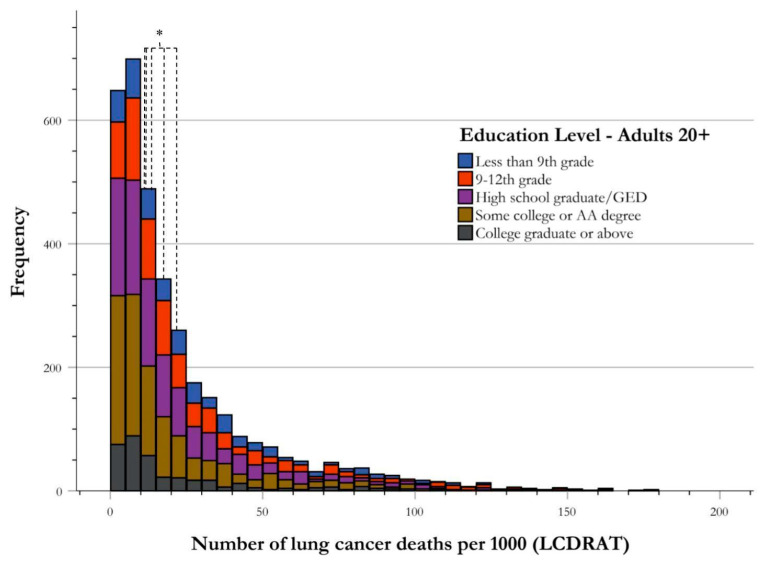
Frequency distribution of risk of lung cancer death by education level. Risk stratification of LC deaths by highest educational attainment level illustrated by frequency of USPSTF-qualified cases of LC deaths per 1000. The central tendency of data for each educational group is established as the median, and is indicated in the figure correlating with group color, with respect to the figure key. The medians of all groups for risk stratification for LC death were found to be significantly different using one-way ANOVA analysis, as shown using an asterisk (*).

**Figure 5 ijerph-21-00781-f005:**
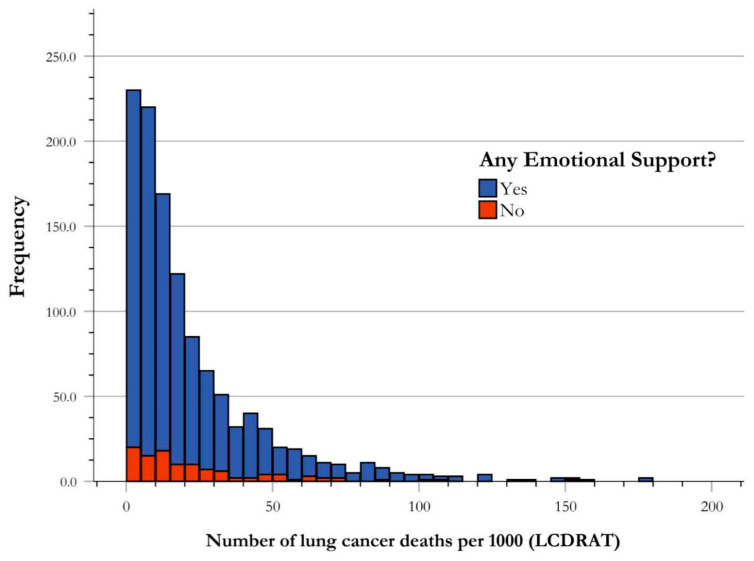
Frequency distribution of risk of lung cancer death by emotional support. Risk stratification of LC deaths by presence or absence of emotional support illustrated by frequency of USPSTF-qualified cases of LC deaths per 1000. The central tendency of data for each emotional support group is established as the median, and is indicated in the figure correlating with group color, with respect to the figure key. The medians of presence and absence of emotional support for risk stratification for LC death were found to not be significantly different using Mann–Whitney U analysis.

**Figure 6 ijerph-21-00781-f006:**
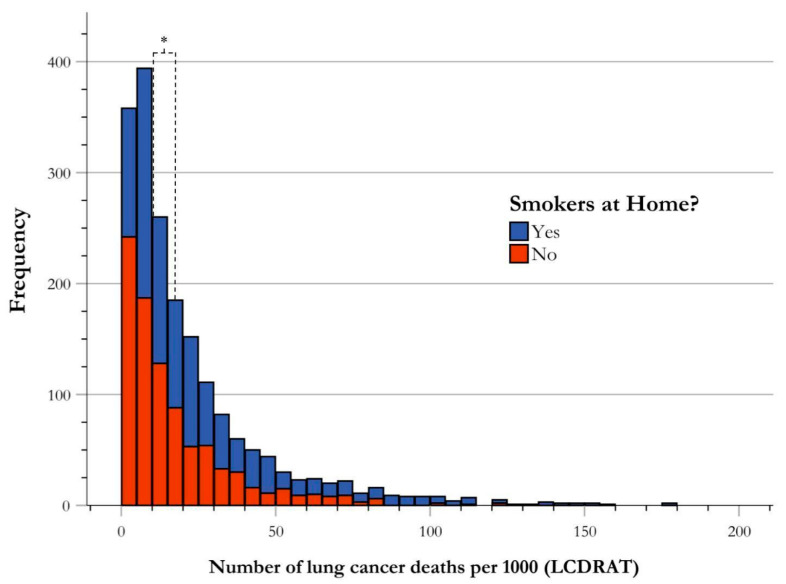
Frequency distribution of risk of lung cancer death by secondhand smoke exposure. Risk stratification of LC deaths by presence of secondhand smoke exposure illustrated by frequency of USPSTF-qualified cases of LC deaths per 1000. The central tendency of data for each group is established as the median, and is indicated in the figure correlating with group color, with respect to the figure key. The medians of all groups for risk stratification for LC death were found to be significantly different using Mann-Whitney U analysis as shown using an asterisk (*).

**Table 1 ijerph-21-00781-t001:** The 2021 USPSTF-eligible lung cancer screening population characteristics from the NHANES (2001–2018).

Characteristic	Number	Percent
Gender		
Male	2271	64.1
Female	1274	35.9
Ethnicity/Race		
African American	801	22.6
Hispanic	463	13.1
Non-Hispanic White	2017	56.9
Other Ethnicity	264	7.4
Educational Attainment		
<9th Grade	440	12.4
9th Grade to 12th Grade	725	20.5
High School Graduate/GED	984	27.8
Some College/AA Degree	1031	29.1
College Graduate and above	364	10.3
Has a history of COPD or Emphysema *	881	24.9
Age in years		
50–60	1204	34
61–70	1389	39.2
71–80	851	24
Body Mass Index (kg/m^2^)		
<20 kg/m^2^	219	6.2
20–25 kg/m^2^	894	25.2
26–30 kg/m^2^	1200	33.9
>31 kg/m^2^	1232	34.8
Smoking History, in pack-years (py)		
20–30 py	994	28
31–50 py	1420	40.1
51–70 py	638	18
71+ py	493	13.9

* COPD or emphysema history was acquired from questions in the medical conditions section of the NHANES database; questions asked included “Have you ever been told that you have COPD?”, “Have you ever been told you have emphysema?”, and “Have you ever been told you have emphysema, COPD, or chronic bronchitis?”.

**Table 2 ijerph-21-00781-t002:** Description of LCDRAT variables from the NHANES (2001–2018) utilized for LCDRAT calculation *.

Variable	Description.
Age	Taken from the age at the time of screening. Variable RIDAGEYR.
Gender	Gender was taken from sex assigned at birth. Variable RIAGENDR.
Ethnicity/Race	Ethnicity/Race was obtained from participants’ asked identity. Variable RIDRETH1.
Educational Attainment	Educational attainment was obtained from participants’ asked education level for adults. Variable DMDEDUC2.
Quit years	Quit years were calculated from time since quit smoking, converted to years. Patients were asked in days, months, or years their total quit time. Variables SMQ852U for units, and SMQ852Q for total time quit.
Smoke Years	Smoke years were calculated from age participants regularly smoked (variable SMD030), subtracted from age at exam (variable RIDAGEYR), subtracted from quit years calculated.
Cigarettes per day	This was obtained from the number of cigarettes actively smoked per day (variable SMD070), or cigarettes smoked when last quit (variable SMD057).
COPD or emphysema history	COPD or emphysema history was acquired from questions in the medical conditions section of the NHANES database; questions asked if participant was ever told they had a history of COPD, emphysema, or chronic bronchitis; compiled from variables MCQ160G, MCQ160K, MCQ160O, and MCQ160P.
Body Mass Index (BMI)	Body mass index in kg/m^2^, taken from the NHANES variable BMXBMI.

* As the NHANES does not ask for a family history of lung cancer, this was a major limitation in the calculation for this study. All participants were assumed to have no genetic risk factors.

**Table 3 ijerph-21-00781-t003:** Pairwise Comparisons of Race and Educational level for number of lung cancer deaths per 1000.

Ethnicity/Race	Significance	Adjusted Significance *
Non-Hispanic White vs. Hispanic	*p* < 0.001	*p* < 0.001
Non-Hispanic White vs. African American	*p* = 0.002	*p* = 0.013
Hispanic vs. Other Race	*p* = 0.951	*p* = 1.000
Education Level		
<9th grade vs. 9th–12th grade	*p* = 0.006	*p* = 0.008
9th–12th grade vs. High School Graduate/GED	*p* < 0.001	*p* < 0.001
High School Graduate/GED vs. Some College or AA Degree	*p* < 0.001	*p* < 0.001
Some College or AA Degree vs. College Graduate and Above	*p* = 0.586	*p* = 0.586

* Significance values (<0.050) for race have been adjusted by Bonferroni correction to control for type 1 errors. Significance values (<0.050) for education have been adjusted using the Benjamini–Hochberg method to control the False Discovery Rate. Asymptotic significance (2-sided tests) are displayed.

## Data Availability

The data presented in this study are available on request from the corresponding author. The data are not publicly available due to the confidential nature of the hospital data.

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
