# Peer review of "Identifying Populations at Risk for Lung Cancer Mortality from the National Health and Nutrition Examination Survey (2001–2018) Using the 2021 USPSTF Screening Guidelines"

_ijerph, 2024, doi:10.3390/ijerph21060781_

Round 1

Reviewer 1 Report

Comments and Suggestions for Authors

Thank you for the opportunity to review this paper on identifying populations at risk for lung cancer mortality. Please find my comments below.

1. In the methods, it was noted that there were 264 NHANES participants who were excluded due to "other race". Is there a reason why these participants could not be examined (was it due to statistical / analytic concerns)? I would think that NHANES would have more detailed race/ethnicity on each participant. Moreover, the LCDART appears to have a race/ethnicity dropdown of "other", so I don't understand why they would be excluded. 

2. Along the same lines, if the LCDART tool was utilized, did authors also enter in BMI, history of lung disease, and family history of lung cancer? I am questioning this in the spirit of ensuring these results can be reproducible. 

3. In the methods or elsewhere, please cite how the question regarding emotional support was asked. As authors have acknowledged, this measure is extremely vague and it's unclear how it has any relevance or utility in this context. NHANES questionnaires are split into modules, so it would be important to know where and how this question has been historically asked.

4. Why is there no Table 1 describing the overall characteristics of the final NHANES population examined? While it's understood that the participants would need to qualify per USPSTF guidelines, a table with the breakdown of their characteristics would again be critical for the reproducibility and understanding of your final analytic dataset.

5. One major component lacking in the discussion and limitations is consider of occupation and environmental exposures. I believe this needs to be addressed. 

Author Response

Dear Reviewer 1, 

Thank you for taking the time to provide such insightful feedback to help improve our manuscript. Please see attachment for our responses to your comments. 

Thank you, 

Vivian Tieu 

Reviewer 2 Report

Comments and Suggestions for Authors

Thank you for the opportunity to review this manuscript. Below are some comments that I believe will improve this article.

1. Introduction

The authors base their work on the use of a tool that identifies the population of patients eligible to undergo LC screening. I would like the authors to describe the screening method in more detail in the introductory part. Are there any National programs related to LC screening in the USA?

2. Discussion

The authors researched the territory of the United States of America. The American population is not covered by universal health insurance. I would like the authors to refer to this important segment in the discussion, which is very important in the context of the availability of health care.

Also, related to the discussion. It would be good if the authors highlighted interventions and examples of good practices that could improve coverage of the population at risk.

Author Response

Dear Reviewer 2, 

Thank you for your foresight and time for improving our manuscript. Please see the attachment for our response to your comments.

Thank you, 

Vivian Tieu 
